# Development of the Anthropometric Grouping Index for the Eastern Caribbean Population Using the Eastern Caribbean Health Outcomes Research Network (ECHORN) Cohort Study Data

**DOI:** 10.3390/ijerph191610415

**Published:** 2022-08-21

**Authors:** Israel A. Almodóvar-Rivera, Rosa V. Rosario-Rosado, Cruz M. Nazario, Johan Hernández-Santiago, Farah A. Ramírez-Marrero, Maxime Nunez, Rohan Maharaj, Peter Adams, Josefa L. Martinez-Brockman, Baylah Tessier-Sherman, Marcella Nunez-Smith

**Affiliations:** 1Department of Mathematical Sciences, University of Puerto Rico at Mayagüez, Mayagüez 00681, Puerto Rico; 2Department of Biostatistics and Epidemiology, Graduate School of Public Health, University of Puerto Rico at Medical Sciences Campus, San Juan 00936, Puerto Rico; 3Department of Exercise Physiology, University of Puerto Rico at Río Piedras, San Juan 00925, Puerto Rico; 4School of Nursing, University of the Virgin Islands, St. Thomas, VI 00802, USA; 5Department of Paraclinical Sciences, University of the West Indies, Saint Augustine, Trinidad and Tobago; 6Department of Family Medicine, Faculty of Medical Sciences, University of the West Indies, Cave Hill BB11000, Barbados; 7Equity Research and Innovation Center, Yale School of Medicine, New Haven, CT 06510, USA; 8Department of Medicine, Yale School of Medicine, New Haven, CT 06510, USA

**Keywords:** anthropometric grouping index, cluster analysis, distribution-free pairwise overlap, ECHORN, Caribbean population

## Abstract

Improving public health initiative requires an accurate anthropometric index that is better suited to a specific community. In this study, the anthropometric grouping index is proposed as a more efficient and discriminatory alternative to the popular BMI for the Eastern Caribbean population. A completely distribution-free cluster analysis was performed to obtain the 11 categories, leading to AGI-11. Further, we studied these groups using novel non-parametric clustering summaries. Finally, two generalized linear mixed models were fitted to assess the association between elevated blood sugar, AGI-11 and BMI. Our results showed that AGI-11 tends to be more sensitive in predicting levels of elevated blood sugar compared to BMI. For instance, individuals identified as obese III according to BMI are (POR: 2.57; 95% CI: (1.68, 3.74)) more likely to have elevated blood sugar levels, while, according to AGI, individuals with similar characteristics are (POR: 3.73; 95% CI: (2.02, 6.86)) more likely to have elevated blood sugar levels. In conclusion, the findings of the current study suggest that AGI-11 could be used as a predictor of high blood sugar levels in this population group. Overall, higher values of anthropometric measures correlated with a higher likelihood of high blood sugar levels after adjusting by sex, age, and family history of diabetes.

## 1. Introduction

It has been documented that obesity is increasing in the Caribbean countries as highlighted in the analysis of 1975–2014 trends of age-standardized mean body mass index (BMI) and age-standardized prevalence of BMI-based obesity in both adult females and males [1]. The value of BMI for population-based research [2,3] and for the formulation of public health-related policy has been generalized [3] using weight and height measures (kg/m^2^) to establish weight and obesity categories in epidemiological studies [4,5,6,7,8]. Nevertheless, controversies surrounding the use of BMI as an obesity index include its inability to consider variation in body fat distribution [3,5,9,10] and its discrepancies in correlating with disease risks in different ethnic groups [11].

In 2015, it was suggested that alternative anthropometric measures should be evaluated to help overcome the limitations of BMI [3]. In recent years, there have been efforts to identify anthropometric indices that can predict disease risk among diverse populations [4,5,10,12,13,14,15,16,17,18,19,20,21,22,23,24,25,26]. Examples of these measures are waist circumference, [12,14,15,17,18,20,21,22,23,24,25,26], hip circumference [14,22], waist-to-hip ratio [12,14,18,21,22,24,25,26], waist-to-height ratio [12,14,21,22,24,25,26], abdominal volume index [23,25], body adiposity index [4,15,16,17,19,21,22,23,24,25], body shape index [13,14,25], visceral adiposity index [17,21,24], percent body fat [19,20], conicity index [21,22,23], ponderal index [22], and body roundness index [10,25].

Using population-based data collected from ethnically diverse population groups of the Eastern Caribbean Region, Zayas-Martínez developed the anthropometric index taking into consideration eight anthropometric measurements: height, weight, waist circumference, hip, biceps, triceps, subscapular and suprailiac skinfolds [27]. A limitation of this index emerged from its construction since some of the measures were correlated with each other. In particular, the suprailiac and subscapular skinfolds were both significantly correlated with waist circumference [28].

In this study, we are presenting the newly developed anthropometric grouping index (AGI) as a more efficient and discriminatory alternative to BMI weight and obesity categories. Given the importance of obesity and diabetes as cardiovascular disease risk factors [29], the correlation between AGI and high blood sugar level was explored to expand the scientific knowledge regarding health characteristics in populations of the Eastern Caribbean.

## 2. Materials and Methods

We conducted a cross-sectional analysis using baseline deidentified data by the Eastern Caribbean Health Outcomes Research Network (ECHORN), which is a cohort study based on Yale University. A description of the ECHORN Cohort Study (ECS) and the sampling procedures was published by Spatz et al. [30]. Briefly, ECS is a population-based prospective cohort study targeting 2961 non-institutionalized adults 40 years and older recruited in communities located in the islands of Barbados, Puerto Rico, Trinidad and Tobago, and United States Virgin Islands between the years 2013 and 2016 [30]. The protocol for the current research was approved by the University of Puerto Rico Medical Sciences Campus Institutional Review Board (protocol number B1940219). We analyzed data from 2891 participants in the referenced study who provided information regarding the diagnosis of high blood sugar (including prediabetes and diabetes) and had available data for the anthropometric measures: weight, height, waist circumference, and hip circumference.

The AGI was constructed using height, weight, waist circumference, and hip circumference data collected during the baseline clinical assessment, as described in the Manual of Procedures of the ECHORN project [31]. Weight was measured in kilograms using a digital scale while the participant was standing up, keeping their head looking straight forward, and their hands hanging at each side. The weight was measured using a digital scale except on patients with clinical contraindications (i.e., participant with a pacemaker). In participants over 200 kg, the weight was estimated using two digital scales. Height was measured in centimeters using a mobile stadiometer while the participant stood up, head straight looking forward, heels together, and toes separated [31].

A tape measure in centimeters was used to obtain both waist circumference and hip circumference in participants without contraindications [31]. Waist circumference was measured after each participant had several consecutive natural breaths while standing. The top of the iliac crest was identified, and the waist circumference was measured midway between the top of the iliac crest and the lower margin of the last palpable rib in the mid axillary line. The procedure was performed twice. If the two measurements differed more than 1 cm, the complete procedure had to be completed again and the first two measurements were discarded [31]. Hip circumference was measured at the largest circumference of the buttocks while each participant was standing. The procedure was also performed twice. If the two measurements differed more than 1 cm, the complete procedure had to be completed again and the first two measurements were discarded [31].

In the present study, the following categories of BMI provided by ECHORN were used: underweight (0, 18.5 kg/m^2^), normal weight [18.5 kg/m^2^, 25 kg/m^2^), overweight [25 kg/m^2^, 30 kg/m^2^), obese I [30 kg/m^2^, 35 kg/m^2^), obese II [35 kg/m^2^, 40 kg/m^2^), and obese III (40, ∞ kg/m^2^). Other study variables were age, sex, and family history of diabetes.

### Statistical Methods

Participants with similar anthropometric characteristics were grouped to form distinguishable clusters (homogeneous groups) using a *k*-means clustering analysis [32]. Groups obtained by *k*-means minimize within-sums-of-squares, i.e., [32]
$$min\;\overset{K}{\underset{k=1}{\sum }}\overset{n}{\underset{i=1}{\sum }}{\left|\left|X_{i}-m_{k}\;\right|\right|}^{2}\zeta_{i}{}_{k};\;$$
where $X_{i}$ is the *i*th participant with the four anthropometric measures, $m_{k}$ is the mean vector associated with the *k*-group and $\zeta_{i}{}_{k}$ is the membership of the *i*-th observation in the *k*-th group. The anthropometric measurements of interest (height, weight, waist circumference, and hip circumference) were standardized. Since the number of categories in AGI was unknown, the number of homogeneous groups was estimated to be from 1 to 15. Eleven categories for AGI were identified using the Jump method of Sugar and James [33] that maximizes the distortion in homogeneous groups.

To further understand the newly obtained categories, a non-parametric estimate of the pairwise overlap, also known as the misclassification probability, of Maitra and Melnykov [34] was computed using Almodóvar-Rivera and Maitra’s approach [35]. The misclassification probability is defined as [34]
$$\omega_{l|k}=P\left(X_{i}\;\mathrm{is}\;\mathrm{assigned}\;\mathrm{to}\;C_{l}|X_{i}\mathrm{was}\;\mathrm{assigned}\;\mathrm{to}\;C_{k}\right).$$

Similarly, $\omega_{k|l}=P\left(X_{i}\;\mathrm{is}\;\mathrm{assigned}\;\mathrm{to}\;C_{k}|X_{i}\mathrm{was}\;\mathrm{assigned}\;\mathrm{to}\;C_{l}\right);$ then, pairwise overlap between the *l*-th and *k*-th group was computed as $\omega_{kl}=\omega_{lk}=\omega_{l|k}+\omega_{k|l}$. The pairwise overlap between two groups is a measure of how distinguishable the groups are from one another. Pairwise overlap values range from 0 to 1 with higher values indicating undistinguishable groups. From these pairwise overlap values, a symmetric overlap matrix Ω is constructed to obtain the corresponding summary measurements. The overlap matrix Ω is defined as
$$\Omega_{K\times K}=\left[\begin{array}{ccc} \begin{array}{cc} 1 & \omega_{21}\\ \omega_{12} & 1 \end{array} & \cdots & \omega_{k1}\\ \vdots & \ddots & \vdots \\ \omega_{1k} & \cdots & \begin{array}{cc} 1 & \omega_{lk}\\ \omega_{kl} & 1 \end{array} \end{array}\right],$$

In summary, these pairwise overlaps are the maximum overlap (the two similar groups) without the diagonal ($\overset{ˇ}{\omega }$), average overlap ($\bar{\omega }$) without the diagonal, and generalized overlap ($\ddot{\omega }$), which is defined as
$$\frac{\lambda_{1}-1}{K-1}$$
where $\lambda_{1}$ is the first eigenvalue of the overlap matrix Ω [34]. Estimates for the pairwise overlaps were carried out using the *R* package *SynClustR* available at the author’s website [35].

To account for the variation among participants between the islands, generalized linear mixed models (GLMMs) were fitted considering the participating Eastern Caribbean islands as random effects. The blood sugar level (dichotomized as normal or high) was considered the dependent variable. The variable was constructed based on responses to questions about ever being diagnosed with pre-diabetes, impaired fasting glucose, impaired glucose tolerance, borderline diabetes, diabetes, or high blood sugar by a doctor or other health professional. One model included the newly proposed AGI with 11 groups as an independent variable; a second model was constructed using BMI. The GLMMs were adjusted by sex, age, and family history of diabetes of parents and grandparents.

## 3. Results

Data from 2891 participants fulfilling the inclusion criteria were considered for analyses. Median age of participants was 57 years, with an interquartile range of 15 years. Over one-third of participants (34.8%) were male, and 26.5% reported high blood sugar levels.

### 3.1. AGI-11

Initially, a *k*-means clustering analysis was performed to obtain homogeneous groups. This approach identified 11 homogeneous groups, referred to as AGI-11. Table 1 presents the estimate of the pairwise overlap for the 11-clustering solution. We must mention that groups are not in order like BMI, but instead based on the groups labelled by the clustering solution. We compared the groups in terms of their misclassification probabilities. Groups 9 and 11 had the highest misclassification probability (${\overset{ˇ}{\omega }}_{9,11}=0.1135$). These two groups have more similar anthropometric characteristics between one another. Other groups that displayed pairwise overlap measures above 0.1 were Groups 7, 8, 9, and 10, meaning that these groups shared some similar anthropometric characteristics between each other. Most misclassification probabilities were below 0.01, indicating that those groups were easily distinguishable from one another. In terms of overall summary, the estimated generalized overlap, also known as the summarized overlap, was $\ddot{\omega }=0.034$. Since the summarized overlap value was less than 0.05, it indicated that most groups were very different from each other [36].

### 3.2. Average Anthropometric Measures by AGI-11 and BMI

Table 2 presents the average and standard deviation of the anthropometric measures of interest (height, weight, waist circumference, and hip circumference) for AGI-11 and BMI. The smallest average values for AGI-11 in the Eastern Caribbean were observed in Group 1. Except for height, all other anthropometric values were the smallest observed (53.2 ± 5.06 kg) for weight, (74.52 ± 5.67 cm) for waist circumference, and (91.8 ± 4.9 cm) for hip circumference. Group 2 included 67 individuals that had the largest average anthropometric values, except for height. The average ± SD was (131.6 ± 13.20 kg) for weight, (126.96 ± 9.98 cm) for waist circumference, and (142.31 ± 7.98 cm) for hip circumference. Group 5 were taller (175.44 ± 5.96 cm), weighed less (111.9 ± 8.78 kg), but had a larger hip measurement (121.83 ± 6.56 cm). Group 7 had one of the smallest average heights (154.8 ± 4.25 cm) and weights (78.18 ± 6.16 kg) but had larger average hip (98.62 ± 5.69 cm) and waist (112.25 ± 5.25 cm) circumferences.

Table 2 also shows the average and standard deviation of anthropometric measures by BMI category. The group with the smallest number of participants was the underweight (*n =* 37) group, representing around 1.28% of the ECS data sample. The group with the highest number of participants was the overweight (*n* = 1024) group, representing 35.42% of the ECS sample. Average height measure was almost similar for all the categories, and it ranged from 162.38 ± 8.66 cm to 166.61 ± 9.41 cm. Average weight increased in each category, with underweight having the smallest value (48.67 ± 6.17 kg) and obese III the largest value (117.46 ± 17.08 kg). Similar to weight measures, waist and hip circumference values increased with each increasing category. Variability for waist and hip circumferences was lower (smaller standard deviations) in the normal weight BMI category group compared with the other groups.

To visualize the categories of AGI-11 and BMI, a parallel coordinates plot was performed. Figure 1 shows all the observations as very clear lines, with thicker lines representing the means of each anthropometric measure for each group. Among the Eastern Caribbean population participating in the ECS, on average, there is a tendency for similar height ranging from 162.38 ± 8.66 cm to 166.61 ± 9.41 cm. One of the most important differences in each group was the weight variable, with significantly increasing values in each group category. Waist and hip circumferences remained constant in each of the BMI categories, providing not much information regarding (Figure 1a) body shape. These results are not surprising since BMI emphasizes in the height and weight of the individual rather than their body shape. Similarly, in Figure 1b, the thicker lines represent the means of each anthropometric measure for each group. Our index can give us information about body shape for each group. Group 10 was represented by the dark blue line, and, on average, their weight was 76.46 ± 6.01 cm compared to 78.18 ± 6.16 cm from those in Group 7 (light purple). However, they differ in hip circumference (86.07 ± 5.71 cm versus 98.62 ± 5.69 cm) and waist circumference (99.17 ± 4.94 cm versus 112.25 ± 5.25 cm).

### 3.3. Association between Blood Sugar Level and AGI Using Generalized Linear Mixed Models (GLMM)

Table 3 presents the estimated prevalence odds ratios (POR) with their corresponding 95% confidence intervals (95% CI) assessing the association between blood sugar level (dichotomized as normal or high) based on the GLMM. Each model was adjusted by age, sex, and family history of diabetes. The first model included AGI-11 as the main independent variable using Group 8 as the reference group. As shown in Table 2, Group 8 is the most similar to the normal category of BMI in terms of average anthropometric measurements.

Compared to ECS participants in Group 8, participants in most other groups (for example, Groups 2, 3, 5, and 7) were significantly more likely to self-report high blood sugar levels. The largest magnitude of association was observed for Group 2 (POR: 3.73, 95% CI: 2.02, 6.86). On average, the anthropometric measures of Group 2 had larger values for weight (131.6 ± 13.2 kg), waist circumference (126.96 ± 9.98 cm), and hip circumference (142.31 ± 7.98 cm). Another important association was observed for Group 5 (POR: 2.66, 95% CI: 1.62, 4.37) who were taller (175.44 ± 5.96 cm), weighed more (111.9 ± 8.78kg), and had higher values of average waist (113.52 ± 7.13 cm) and hip (121.83 ± 6.56 cm) circumferences as compared to Group 8. The comparison with Group 7 reflected a smaller height (154.8 ± 4.25 cm), but larger average values for weight (78.18 ± 6.16 kg), waist circumference (98.62 ± 5.69 cm), and hip circumference (112.25 ± 5.25 cm). As compared to the reference group, Group 7 was also more likely to self-report high blood sugar levels (POR: 2.37, 95% CI: 1.54, 3.64). The second model was constructed using BMI as the main independent variable. As expected, the results suggest a dose–response relationship between the BMI categories and the likelihood of self-reporting high blood sugar levels. The strongest association was observed in the obese III category (POR: 2.57, 95% CI: 1.68, 3.74).

## 4. Discussion

The current cross-sectional analysis used ECS baseline population-based data for 2891 participants living in the Eastern Caribbean region. The availability of anthropometric measurements of weight, height, waist circumference, and hip circumference made the construction of the AGI possible. A complete unsupervised approach not only allows us to identify these categories but also to study them in terms of characteristics between groups. These anthropometric measures allowed the description of the following characteristics for each of the AGI-11 groups: (1) anthropometric measures of weight and height have less variability; and (2) waist and hip circumferences and their variation in the Caribbean population. More importantly, our results showed that AGI-11 tends to be more sensitive in predicting levels of elevated blood sugar as compared to BMI. For instance, individuals identified as obese III according to BMI are (POR: 2.57; 95% CI: (1.68, 3.74)) more likely to have elevated blood sugar levels, while, according to the AGI, individuals with similar characteristics are (POR: 3.73; 95% CI: (2.02, 6.86)) more likely to have elevated blood sugar levels. Participants with the larger average values for both waist and hip circumferences had the higher likelihood of reporting high blood sugar levels (groups 2, 3, and 5) as compared to Group 8. An app where users can input their anthropometric measures to see which AGI category, they belong to will be available at www.echorn.org (accessed on 12 July 2022).

We feel that this work is an important contribution and that it will motivate other researchers to further explore these issues. For instance, waist and hip circumferences can also be used in future studies to detect specific body morphologies, such as “apple-shaped body” (larger fat distribution around the waist) and “pear-shaped body” (larger fat distribution around the hips), that have been associated with metabolic risk. In this work, we estimated the number of categories to obtain the most similar groups. However, methods such as *k*-means can leave the choice of desired categories to the researcher. It might be possible to combine groups who share similar characteristics to create heterogeneous groups, leading to a smaller generalized overlap as suggested by Almodóvar-Rivera and Maitra [35]. A limitation of this study was the lack of information available regarding other anthropometric measurements, such as skinfolds and arm circumference. Additionally, data about low sugar levels were not available.

## 5. Conclusions

In conclusion, the findings of the current study suggest AGI-11 could be used as a predictor of high blood sugar level in the ECS population group. Overall, GLMMs presented results that correlated higher values of anthropometric measures with a higher likelihood of high blood sugar levels. For instance, individuals with the anthropometric for weight (131.6 ± 13.2 kg), waist circumference (126.96 ± 9.98 cm), and hip circumference (142.31 ± 7.98 cm) were 3.73 times more likely to have elevated blood sugar levels.

## Figures and Tables

**Figure 1 ijerph-19-10415-f001:**
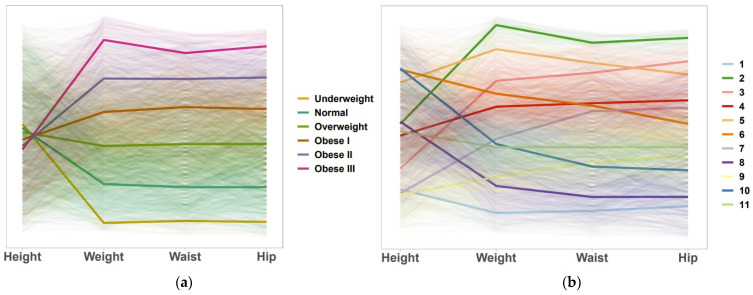
Parallel coordinates plot for categories of BMI and AGI-11 under the four anthropometric measurements (height, weight, waist circumference, hip circumference): (**a**) BMI; (**b**) AGI-11.

**Table 1 ijerph-19-10415-t001:** Pairwise overlap matrix for the 11 categories for AGI.

	1	2	3	4	5	6	7	8	9	10	11
1	1	0.0001	0.0001	0.0003	0.0001	0.0002	0.0011	0.0971	0.0902	0.001	0.0076
2	0.0001	1	0.0026	0.0002	0.0059	0.0001	0.0001	0.0001	0.0001	0.0001	0.0001
3	0.0001	0.0026	1	0.0652	0.0185	0.0022	0.0319	0.0001	0.0005	0.0002	0.0014
4	0.0003	0.0002	0.0652	1	0.0121	0.0705	0.1117	0.0013	0.0058	0.0073	0.0866
5	0.0001	0.0059	0.0185	0.0121	1	0.0238	0.0007	0.0001	0.0001	0.0003	0.0005
6	0.0002	0.0001	0.0022	0.0705	0.0238	1	0.0026	0.0012	0.0007	0.0573	0.0308
7	0.0011	0.0001	0.0319	0.1117	0.0007	0.0026	1	0.0021	0.0786	0.0013	0.0796
8	0.0971	0.0001	0.0001	0.0013	0.0001	0.0012	0.0021	1	0.0669	0.0594	0.1108
9	0.0902	0.0001	0.0005	0.0058	0.0001	0.0007	0.0786	0.0669	1	0.0028	0.1135
10	0.001	0.0001	0.0002	0.0073	0.0003	0.0573	0.0013	0.0594	0.0028	1	0.1133
11	0.0076	0.0001	0.0014	0.0866	0.0005	0.0308	0.0796	0.1108	0.1135	0.1133	1

**Table 2 ijerph-19-10415-t002:** Summary of anthropometric measurements for AGI-11 and BMI.

	*K*	*n_k_*	Height (cm)	Weight (kg)	Waist (cm)	Hip (cm)
AGI-11	1	223	155.38 ± 4.44	53.2 ± 5.06	74.52 ± 5.67	91.8 ± 4.9
	2	67	166.92 ± 7.97	131.6 ± 13.20	126.96 ± 9.98	142.31 ± 7.98
	3	192	159 ± 4.46	98.10 ± 7.77	109.52 ± 7.76	126.6 ± 5.39
	4	347	165.1 ± 3.48	89.09 ± 6.63	100.38 ± 6.23	114.03 ± 5.05
	5	134	175.44 ± 5.96	111.9 ± 8.78	113.52 ± 7.13	121.83 ± 6.56
	6	269	177.67 ± 4.87	93.47 ± 7.04	100 ± 5.81	108.39 ± 5.07
	7	270	154.8 ± 4.25	78.18 ± 6.16	98.62 ± 5.69	112.25 ± 5.25
	8	286	167.3 ± 4.41	62.46 ± 5.5	78.5 ± 5.61	94.05 ± 5.19
	9	373	154.55 ± 4.34	65.46 ± 5.4	86.63 ± 5.45	101.78 ± 4.51
	10	259	177.75 ± 4.57	76.46 ± 6.01	86.07 ± 5.71	99.17 ± 4.94
	11	471	165.59 ± 3.42	75.41 ± 5.80	90.43 ± 5.24	103.38 ± 4.81
BMI	1: Underweight	37	166.61 ± 9.41	48.67 ± 6.17	69.89 ± 7.24	87.84 ± 7.24
	2: Normal	707	165.93 ± 9.68	62.64 ± 8.44	80.61 ± 7.13	95.64 ± 5.71
	3: Overweight	1024	165.26 ± 9.45	75.46 ± 9.39	90.89 ± 7.42	103.95 ± 5.87
	4: Obese I	659	164 ± 9.21	86.89 ± 10.41	99.33 ± 8.15	111.74 ± 6.73
	5: Obese II	313	162.88 ± 8.44	98.73 ± 11.10	107.02 ± 9.14	120.44 ± 7.26
	6: Obese III	151	162.38 ± 8.66	117.46 ± 17.08	117.65 ± 12	134.18 ± 10.51

**Table 3 ijerph-19-10415-t003:** Prevalence odds ratios ^a^ and 95% confidence intervals for AGI-11 and BMI to predict reported elevated sugar levels.

	*K*	POR	95% CI
AGI-11	1	1.65	(1.05, 2.61)
	2	3.73	(2.02, 6.86)
	3	2.79	(1.76, 4.41)
	4	1.97	(1.31, 2.98)
	5	2.66	(1.62, 4.37)
	6	2.23	(1.45, 3.44)
	7	2.37	(1.54, 3.64)
	8 ^b^	1	-
	9	1.51	(0.99, 2.29)
	10	0.93	(0.57, 1.51)
	11	1.81	(1.23, 2.66)
BMI	1: Underweight	0.2	(0.05, 0.85)
	2: Normal ^b^	1	-
	3: Overweight	1.17	(0.92, 1.48)
	4: Obese I	1.87	(1.45, 2.42)
	5: Obese II	2.01	(1.47, 2.75)
	6: Obese III	2.57	(1.68, 3.74)

^a^ Adjusted by sex, age and family history of diabetes. ^b^ Index used as a reference.

## Data Availability

The data underlying this article will be shared upon reasonable request to the ECHORN Data Access and Scientific Review committee. Please see https://www.echorn.org/request-echorn-data (accessed on 12 July 2022) for further information regarding data inquiries.

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
