# Peer review of "Development of the Anthropometric Grouping Index for the Eastern Caribbean Population Using the Eastern Caribbean Health Outcomes Research Network (ECHORN) Cohort Study Data"

_ijerph, 2022, doi:10.3390/ijerph191610415_

Round 1

Reviewer 1 Report

Abstract is not fluent. For example, the use of AGI (really CAI?) and AGI-11 randomly should be revised. Also, the clear aim of present study is lacking. As a reader the main aspects of manuscript should be presented not what cluster analyses is… these can be opened in introduction?

Introduction ok

Methods: For AGI-11, the groups and grouping should be more accurately presented

Results- figure 1 legend should be more conclusive. Explanation for each variable written out especially Fig A BMI. Also, the quality of graph (fig 1A and B) is not sufficient.

To be ones the BMI seem to be more tempting measure after reading the results. As authors state the order for AGI-11 does not follow. As authors state groups are not in order. It is fair to as a reader why not? The more easy and logical manuscript and presentation is the more it has to offer. The mathematics are already really complex on present article to help reader please but groups in order while presenting results and as mentioned above the each group should be clearly presented (cut-off values…) if not in manuscript then an appendix?

Discussion is preliminary?

Conclusion- see comments above. Higher values correlate with higher risk for high blood sugar levels. This should be clearly presented in the manuscript in logical manner.

Reviewer 2 Report

This work attempts to test a clustering method that best fits the detection of diabetes, assessed by patient opinion. However, this grouping is more complex due to the number of clusters, which makes it more impractical and difficult to understand. The authors must test different methods, algorithms, and groupings, among them the k-medoids and the k with six components (AGI-6); likewise, demonstrate that the proposed grouping (AGI-11) is more sensitive than the BMI-6 or BMI-11. It is somewhat logical that the greater the number of components, the greater the method's sensitivity.

Lines 65-66: What here is mentioned it is not a limitation. The correlations are necessary; otherwise, the studies would not be carried out. The problem is the intensity of the collinearity.

Methods

Why were k-means used instead of K-medoids? How was the AGI-11 model validated concerning other clusters? What did you do with the extreme values? Mention how many iterations were performed to calculate the 11 centroids in each analysis method (100 are recommended). If necessary, use bootstrapping to perform 1000 iterations, and test that the computation of centroids stabilized. Give the value of the Jaccard coefficient. Eleven centroids complicate understanding. Try other metrics or algorithms than Sugar James and stick with the simplest, most sensitive, and most informative K-n.

Results

Table 2:

To facilitate the comparison between each method, and its subsequent discussion, add the BMI classification for each AGI-11 cluster.

Add a six-component analysis similar to BMI by the AGI method (AGI-6).

As in the upper part of the AGI-11 table, add the BMI cluster number: 1 underweight, 2 normal weight...

Add a similar table as an attached file, ordering the data according to the average BMI found for each cluster.

Figure 1a, 1b: Improve the quality of the figure.

Table 3:

Add the same as requested in Table 2.

It is somewhat logical that the greater the number of clusters, the greater the prediction sensitivity. However, the greater the number of clusters, the more difficult it is to understand. As in table 2, add the AGI-6 grouping.

I do not observe the higher sensitivity mentioned since the confidence intervals are double in AGI-11 than in BMI-6.

Add a similar table as an attached file, ordering the data according to the OR.

Lines 250-251: incomplete idea.

Lines 269-270: For this particular work, what here is mentioned is not a limitation.

Round 2

Reviewer 1 Report

Please see file 4.8.2022 for detailed comments. Manuscript has improved but there are some issues that should be addressed. At points manuscript is hard to follow or lacks accurate information.

Reviewer 2 Report

Thanks to the authors for the clarifications and for heeding the recommendations.
